# Match Load Physical Demands in U-19 Professional Soccer Players Assessed by a Wearable Inertial Sensor

**DOI:** 10.3390/jfmk8010022

**Published:** 2023-02-07

**Authors:** Guglielmo Pillitteri, Valerio Giustino, Marco Petrucci, Alessio Rossi, Ignazio Leale, Marianna Bellafiore, Ewan Thomas, Angelo Iovane, Antonio Palma, Giuseppe Battaglia

**Affiliations:** 1Sport and Exercise Sciences Research Unit, Department of Psychology, Educational Science and Human Movement, University of Palermo, 90144 Palermo, Italy; 2Palermo FC, 90046 Palermo, Italy; 3Department of Computer Science, University of Pisa, 56127 Pisa, Italy; 4National Research Council (CNR), Institute of Information Science and Technologies (ISTI), 56124 Pisa, Italy; 5Regional Sports School of CONI Sicilia, 90141 Palermo, Italy

**Keywords:** soccer, sport performance, inertial sensor, inertial sensor device, inertial measurement unit, training load, external load, physical demand

## Abstract

Background: Wearable inertial sensors are poorly used in soccer to monitor external load (EL) indicators. However, these devices could be useful for improving sports performance and potentially reducing the risk of injury. The aim of this study was to investigate the EL indicators (i.e., cinematic, mechanical, and metabolic) differences between playing positions (i.e., central backs, external strikers, fullbacks, midfielders, and wide midfielder) during the first half time of four official matches (OMs). Methods: 13 young professional soccer players (Under-19; age: 18.5 ± 0.4 years; height: 177 ± 6 cm; weight: 67 ± 4.8 kg) were monitored through a wearable inertial sensor (TalentPlayers TPDev, firmware version 1.3) during the season 2021–2022. Participants’ EL indicators were recorded during the first half time of four OMs. Results: significant differences were detected in all the EL indicators between playing positions except for two of them (i.e., distance traveled in the various metabolic power zones (<10 w) and the number of direction changes to the right >30° and with speed >2 m). Pairwise comparisons showed differences in EL indicators between playing positions. Conclusions: Young professional soccer players showed different loads and performances during OMs in relation to playing positions. Coaches should consider the different physical demands related to playing positions in order to design the most appropriate training program.

## 1. Introduction

Soccer is a situational team sport in which players are interconnected in a complex system characterized by technical–tactical components that are supported by physical and physiological factors [1]. In soccer, the physical performance consists of intermittent cyclic and acyclic activities characterized by aerobic and anaerobic demands [2,3]. The need to know the physical demands required during a soccer match is of fundamental importance for coaches and athletic trainers in order to properly plan the training program with the aim of increasing the training effect and reducing the risk of injury [4,5,6,7].

High-speed running, acceleration, and deceleration characteristics are considered determinant factors for physical performance and should be taken into consideration when designing the training program [8]. As a matter of fact, players usually perform 150–250 different actions and 1100 changes of direction during a match [9]. Moreover, players’ physical activity characteristics change every 4–6 s based on the different player’s positions on the pitch [9,10], resulting in a high rate of change in speed (i.e., acceleration). Studies have shown that professional players travel a total distance of between 10 and 13 km during a match [11]. Most of the total distance is covered at low intensities, whereas 22–24% is spent at intensities above 15 km/h, 8–9% above 20 km/h, and 2–3% above 25 km/h. Additionally, it has been found that players can perform between 600 and 650 accelerations during a match.

To improve performance while reducing the risk of injury, practitioners should assess match loads in relation to playing positions on the pitch to optimize training planning [12,13]. Specifically, the external load (EL) is usually described by the total distance, range of speed covered, accelerations, metabolic power [14], and other derived measures. Global position systems (GPS) technology has been largely used by practitioners to assess EL allowing for time-motion analysis in technical–tactical tasks [6,15,16,17]. However, in the last few years, several microelectromechanical systems (MEMS) have been developed and are available and these include triaxial accelerometers, triaxial gyroscopes, magnetometers, and pressure sensors. These devices, defined as inertial sensors devices (ISDs) or inertial measurement units (IMUs), can measure acceleration and angular velocity, among other parameters. ISD technology was developed specifically for the assessment in indoor sports where GPS devices cannot be used [18]. The literature suggests that this technology has not yet been fully used in professional soccer for EL monitoring.

It should be mentioned that physical demand is highly related to playing positions on the pitch due to the fact that roles have specific technical–tactical requests strictly related to different physical, physiological, energetic, and biomechanical components [2,13,19,20]. For example, the literature has reported that the total high-intensity distance is covered by central midfielders, wide defenders, and wide midfielders, whereas strikers and central defenders travel lower distances [21]. Moreover, wide defenders and wide midfielders perform the highest sprinting distance, whereas central midfielders and central backs cover the lowest in elite soccer players [21,22,23,24,25]. Moreover, significant differences were found between the various playing positions for all measures of EL in amateur soccer players [26]. Authors reported that central midfielders covered the longest distance during a match, which is in line with recent literature followed by the forwards, the full-backs, and the wide midfielders. As in adult professional soccer players, physical performance is affected by playing position also in youth elite players. In young male elite players (8–18 years), center backs covered the shortest high-intensity running and sprinting distances and wide midfielders the longest [27]. Additionally, in young soccer players (mean age 16.0 years), center defenders covered the shortest very high-speed (≥19.8 km·h^−1^) and sprint (≥25.2 km·h^−1^) distances, whereas wide players and center forwards covered the longest distances in these speed zones [28], which is in line with other studies [29].

It is worth noting that the GPS may have some signal issues due to adverse weather conditions or being used indoors compared to ISD. In fact, being that this technology is based on (inertial) movement, it allows for evaluating the EL considering the same parameters relating to distance, speed, and metabolic power with greater precision than GPS at 10 Hz [30]. Indeed, the better applicability of ISD technology is due to its small size, lower cost, and the possibility of using the device indoors, avoiding possible connection problems between GPS and satellite [30].

To the best of our knowledge, there are no studies that have investigated, through ISD technology, differences in EL indicators in young professional soccer players during official matches (OMs) considering the different playing positions on the pitch. We hypothesized that there may be differences in EL indicators in young professional soccer players during OMs depending on playing position. Hence, the aim of this study was to investigate any differences in EL indicators, specifically cinematic, mechanical, and metabolic indicators, measured through a wearable inertial sensor during the first half time of four OMs, considering the different playing positions in young professional soccer players (U19).

## 2. Materials and Methods

### 2.1. Study Design

In this cross-sectional study, young players from a professional Italian soccer club were monitored using a wearable inertial sensor during the first half time of four OMs during the season 2021–2022. Soccer players were categorized into five groups according to their playing position on the pitch as follows: central back (CB), external striker (ES), fullback (FB), midfielder (MD), and wide midfielder (WM).

### 2.2. Participants

Thirteen young professional soccer players (age: 18.5 ± 0.4 years; height: 177 ± 6 cm; weight: 67 ± 4.8 kg) competing in the Italian U19 Championship were included. Participants’ playing positions were the following: CB (*n* = 2), ES (*n* = 3), FB (*n* = 5), MD (*n* = 1), and WM (*n* = 2). The following inclusion criteria were considered: (1) professional male soccer players belonging to the Under-19; (2) no injury in the previous six months. Based on the exclusion criteria, only goalkeepers were not eligible for the study. 

All participants signed an informed consent form before taking part in the study. The study, which complies with the principles of the Declaration of Helsinki, was approved by the Bioethics Committee of the University of Palermo (n. 68/2021).

### 2.3. Procedures 

Participants were monitored during four OMs in a regular pitch with a theoretical match density (m^2^/player) referred to ~320 m^2^ according to Riboli et al. [31,32]. OMs were played on a third-generation artificial pitch or natural grass. All participants performed a typical 25-min pre-match warm-up before each OM. 

The ISD was started 5 min prior to the assessment. Data were collected through a wearable inertial sensor (TalentPlayers TPDev, firmware version 1.3) [30]. To avoid interunit errors, each participant was assigned the same ISD for each OM. 

Among the available devices, TalentPlayers was chosen as it provides very similar data compared to traditional GPS systems (i.e., instantaneous speed and distance, change of directions, and metabolic data) and it is already used by various Italian soccer teams (https://talentplayers.com (accessed on 16 December 2016)). This ISD is a small wearable device integrating a six degrees of freedom MEMS inertial sensor, capable of providing real-time acceleration and rotation data along three orthogonal axes at a frequency of 100 Hz per channel. It is designed to be worn on the lower leg using an elastic band. The validity and reliability of the ISD have been previously reported [30].

All data were acquired by the TalentPlayer mobile app (software version 1.0.7) and uploaded to the TalentPlayers cloud.

EL indicators considered for this study were classified as cinematic, mechanical, and metabolic. All indicators assess the volume of OMs (except the metabolic power indicator) although some parameters represent intensity performance indicators. These are detailed in Table 1.

### 2.4. Statistical Analysis 

Normality distribution was calculated through the Shapiro–Wilk test. Means and standard deviations of all the EL indicators for OMs and for each playing position (CB, ES, FB, MD, WM) were provided. 

A one-way analysis of variance (ANOVA) test on one factor (OM) was performed to detect differences for each EL indicator. The Tukey post hoc test was used for pairwise comparisons for each EL indicator between playing positions. Statistical significance was set at *p* < 0.05.

The Statistical Package jamovi (The jamovi project—jamovi Version 1.8.0.1) was used to perform data analysis. Graphs were created through Graph Pad Prism 8 (Version 8.0.2).

## 3. Results

Descriptive statistics of EL indicators (i.e., TD, MS, N°INTACC, N°INTDEC, TDA, TDD, N°HSR, WT, THSR, WD, DHSR, MP, TLMP, THMP, TEMP, TMMP, DLMP, DHMP, DEMP, DMMP, N°CoDR, N°CoDL) for each playing position (i.e., CB, ES, FB, MD, WM) are reported in Table 2, Table 3, Table 4 and Table 5.

As reported in Table 6, results from the one-way ANOVA tests showed significant differences in all the EL indicators between playing positions (TD: F(4,13,9) = 16.59, *p* < *0*.001; MS: F(4,11,8) = 5.54, *p* = 0.009; N°INTACC: F(4,12,5) = 6.22, *p* = 0.005; N°INTDEC: F(4,11,7) = 3.43, *p* = 0.045; TDA: F(4,14,2) = 9.99, *p* < 0.001; TDD: F(4,13,7) = 23.29, *p* < 0.001; N°HSR: F(4,13,7) = 6.94, *p* = 0.003; WT: F(4,13,1) = 5.21, *p* = 0.01; THSR: F(4,14) = 7.82, *p* = 0.002; WD: F(4,14) = 4.76, *p* = 0.012; DHSR: F(4,14) = 7.89, *p* = 0.002; MP: F(4,14,2) = 8.83, *p* < 0.001; TLMP: F(4,12,6) = 6.22, *p* = 0.005; THMP: F(4,14,1) = 18.75, *p* < 0.001; TEMP: F(4,12,2) = 4.1, *p* = 0.025; TMMP: F(4,13,1) = 7.04, *p* = 0.003; DHMP: F(4,14,1) = 15.51, *p* < 0.001; DEMP: F(4,12,3) = 5.00, *p* = 0.013; DMMP: F(4,13,2) = 7.6, *p* = 0.002; N°CoDL: F(4,14) = 13.03, *p* < 0.001) except for DLMP: F(4,12,4) = 1.43, *p* = 0.28 and N°CoDR: F(4,13,9) = 2.93, *p* = 0.06.

Details of the Tukey post hoc analysis reporting the pairwise comparisons between playing positions are provided in Table 6 and Table 7. The data show that all the EL indicators differ for each playing position except for N°INTDEC, WD, DLMP, and N°CoDL. 

Figure 1, Figure 2 and Figure 3 show cinematic, mechanical, and metabolic indicators performed during OMs, respectively.

## 4. Discussion

The aim of this study was to investigate any differences in EL indicators during the first half time of four OMs between young professional soccer players based on their playing position. The EL indicators were assessed using a wearable inertial sensor device. This technology allows us to assess EL considering several indicators similar to GPS avoiding signal issues and connection problems. Moreover, Coutts et al. [35] have demonstrated that GPS devices are reliable in assessing total distance and peak speed during high-intensity intermittent exercise but are less reliable for high-intensity activities such as accelerations. In contrast, the ISD device tracks activity information using inertial technology that captures acceleration and rotation data in real time at a rate of 100 Hz per channel, resulting in more accuracy for detecting acceleration and high-intensity effort than GPS.

As we hypothesized, the main findings of our study showed significant differences in EL indicators between playing positions during OMs.

The scientific literature shows that physical demand in soccer players has been largely studied using GPS technology [17,36,37]. The main studies that assessed external load performance during matches found that physical performance is highly specific according to the role of the players. These results are in line with our findings, although we measured the EL indicators through an ISD. Indeed, considering cinematic, mechanical, and metabolic external load indicators, the ES, WM, and MD performed the highest level of physical performance during the first half time of the matches, whereas the CB and FB had the lowest level. 

### 4.1. Cinematic External Load Indicators

In our study, significant differences in cinematic EL indicators between playing positions during the first half time of an OM were detected. Several previous studies have found differences in EL as distance, speed, and accelerations [38,39,40] between playing positions during both training and competition including both elite and amateur soccer players [26]. Moreover, considering physiological characteristics (i.e., HR and derived indices) [41,42] similar results have been found. The literature reports that the longest distance covered at high intensity has been achieved by the WM and FB [40,41,42,43]. In our study, we found that the WM and MD covered the highest TD during the first half time of an OM (6161 ± 316 and 5963 ± 74 m, respectively), whereas the CB (5240 ± 340 m) and FB (5087 ± 344 m) the lowest. Our results are in agreement with Ingebrigtsen et al., (2014) in which authors reported that, during the first half time of the match, WM traveled the highest TD and CB the lowest [44]. 

Concerning intensity indicators (i.e., N°HSR, THSR, and DHSR), ES and WM showed the highest results, whereas CB and FB showed the lowest. In line with these results, we found that FB and CB showed the highest amount of WT whereas MD and WM the lowest. However, there was no significant difference between the roles for the WD even though the FB and ES showed highest and the MD and CB the lowest values. Studies reported that CB showed the longest recoveries between consecutive high-intensity efforts [45] and spent the most time in low intensity efforts [39], in line with our results. Moreover, the MD spent less time in very low activity [46] and stand for much less time than other playing positions [47]. Indeed, MDs perform low to moderate-intensity activity more frequently showing shorter recovery bouts between high intensity efforts [46,47]. 

It is worth noting that when speed intensity increases, the MD exhibits low results. In fact, MDs play in very dense central spaces that limit the performance of intense actions such as high-speed running. However, when expressed as metabolic power, the central MD showed higher volume of high-intensity activity compared to attackers due to the accelerations [48].

The literature reports that FB perform more high intensity running than other playing positions. Specifically, Bangsbo et al., reported that MDs, FBs, and attackers covered a greater distance in high intensity running than the defenders [47,48,49]. Our results indicated that FB performed less intensity activity than other playing positions, showing similar performance to CB. Probably, in our sample, FB were required to have more defensive tactical function (i.e., 4:3:3 system) than offensive ones that requires a more intense effort during an OM. Indeed, some contextual factors such as tactics, game location, opponent quality, congested period, or match status could influence physical performance [50,51]. A study carried out by Altmann et al., (2021) [52] have demonstrated that FB (e.g., 4:4:2 or 4:2:3:1 system) displayed lower total and high-intensity distances compared to FB (e.g., 5:3:2 system), which is a new finding that emphasizes the need of differentiating between these two positions based on tactical system used. Moreover, WM, FB (in 3:5:2), and FB followed by forwards showed the greatest sprinting distance, whereas MD and CB showed shorter distances while sprinting. These findings are generally supported by previous literature [21,22,24,25].

Previous research indicates that FB, attackers, and MD players (both central and wide) covered the highest amounts of HSR and sprinting distance [25,53]. Dalen et al., (2016) reported that the FB and WM covered the highest whereas the CB the lowest HSR distance in an OM [54]. Ingebrigtsen et al., (2014) showed a greater HSR for the WM and FB and less for the CB in the first half of the match [44]. Additionally, authors reported the same results considering sprinting and distance [44]. In the same way, Dalen et al., (2016) reported highest sprinting distance for the FB and WM and less for the CB [54]. Moreover, Oliva Lozano et al., (2020) detected a sprinting distance greater for the WM and lower for the MD during an OM [8].

A high requirement for such running patterns in attacking players (e.g., WM) it may be necessary to cope with tactical demands related to overcoming defensive strategies to set up scoring situations. The MD plays in dense zones of the pitch that limit sprinting (i.e., speed >25.2 km/h) performance.

We also found that the players who reached the highest speed peaks were the ES and the WM, whereas CB and FB the lowest. These findings are in contrast with a previous study carried out by Rampinini et al., (2007) that reported a peak speed significantly higher for fullbacks than central backs during an OM [53]. Our results are in line with Oliva Lozano et al., (2020) [8] that indicate the highest peak speed for the WM and the lowest for the MD. The bigger space available for the WM compared to the CB and MD can explain these results. 

### 4.2. Mechanical External Load Indicators

A determining factor in soccer performance is the acceleration profile [44,55]. Given the rate of change in speed performed by the players [56], the acceleration profile can be considered as a group of acceleration-based variables that requires a high neuromuscular physical demand [57]. Indeed, high-intensity accelerations and decelerations have a considerable impact on soccer players’ mechanical load and can be counted as markers of muscle damage post-match [58]. Specifically, accelerations have a high metabolic cost [59], whereas decelerations increase the mechanical load [54]. 

Oliva Lozano et al., (2020) [8] have considered high intensity (i.e., >3 m/s^2^) and low intensity (i.e., <3 m/s^2^) accelerations. Authors reported that WM performed the highest and CB the lowest number of intense accelerations during an OM. However, different results have been found with a lower intensity threshold (i.e., low intensity accelerations). Indeed, the MD achieved the highest and WM the lowest number of accelerations. This result may be explained by the fact that density increases (reduced m^2^ per player) as the ball is closer to the central zones of the pitch in match play resulting in a decrease in the intensity of play [32,60].

In our study, ES and FB performed a greater number of intense accelerations than other playing positions whereas the MD and CB are the lowest. The literature reported that in the first half of an OM, the highest number of accelerations have been showed by WM whereas the CB showed the lowest [54]. In this way, Ingebrigtsen et al., (2014) detected similar results. Authors reported that the WM and FB demonstrated a higher amount than the MD and CB [44].

Furthermore, our results confirm the findings of previous research which reported that the WM performed a higher number of intense accelerations than the CB [8].

Although the ES and WM showed the highest number of decelerations, there is no significant difference between the roles in our study.

Our results are in line with previous research where was reported that the WM performed higher number of intense decelerations than the CB [8]. Moreover, Oliva Lozano et al., (2020) found that the WM performed the highest number of intense decelerations and the CB the lowest during an OM, whereas the MD attained the highest and the WM the lowest number of low decelerations [8]. Moreover, Dalen et al., (2016) reported that the FB and WM performed a higher number of decelerations than the CB in the first half time of an OM [54].

We found that the WM and ES covered the highest TDA and FB and CB the lowest. This means that the FB performed shorter accelerations, similar to the CB, compared to the ES and WM. 

In general, previous studies revealed that players in wide positions accelerated significantly more than central players [38,54]. Indeed, Oliva-Lozano et al., (2020) found that players covered longer acceleration distances in external positions (i.e., WM and FB) than the central MD and CB [8]. It is worth noting that Dalen et al., (2016) reported that the FB and WM covered the highest acceleration distance in the first half of the match whereas the CB was the lowest [54]. In addition, Abbott et al., (2018) found the highest intensity acceleration distances in wide positions (i.e., attackers and wide defenders) producing the highest distances due to the frequent tactical requirement of wide positions to reach high speeds [38]. 

In our study, whereas the MD and WM showed the highest decelerations distance, the FB and CB recorded the lowest. This result is in contrast with Dalen et al., (2016) in which a higher deceleration distance for CB and FB compared to WM in the first half of an OM was found [54]. However, there is a difference between the acceleration and deceleration thresholds used between the studies that do not allow us to compare them. Additionally, Oliva Lozano et al., (2020) reported that WM and forward covered the highest deceleration distance whereas the MD and CB were the lowest, accordingly to our results (except for the MD) [8]. However, different methods, MEMS technology, and thresholds used for classifying accelerations and decelerations make it difficult to draw conclusions about this difference.

In field-based intermittent sports such as soccer, it is proposed that the ability to execute rapid changes in direction is a critical factor in relation to match outcomes [61]. Therefore, the ability to make a quick change of direction is related to the ability to produce a large amount of force in a short time [62].

Research has shown that soccer players undertake approximately 700 direction changes of varying intensity during a match, and 600 of these changes in direction are 0–90° turns [2]. Around 50 of the direction changes are performed at maximal intensity during a match [2]. Approximately 700 direction changes per match were made by defenders, 500 by midfielders, and 600 by strikers. However, midfielders and strikers performed more turns of 270° to 360°. This could be due to specific tactical requirements such as playing position in midfield. The amount of 90° to 180° turns is relatively uniformly distributed with all positions performing roughly between 90 and 100 in official matches [2]. In our study, we considered a change of direction >30° and performed at a speed >2 m/s, classifying them in right and left changes of direction.

Regarding N°CoDR, we found that the difference appears to be significant only between the FB and WM. Specifically, the WM and ES performed the highest amount of CoDR, whereas the FB and MD the lowest. It is worth noting that, considering N°CoDL, there is no significant difference between the roles, even if the MD and CB performed the highest number whereas ES and WM were the lowest.

### 4.3. Metabolic External Load Indicators

Metabolic power, defined as the product of the energetic cost of acceleration running (EC, J·kg^−1^·m^−1^) and speed (v, m·s^−1^), has been considered for EL monitoring as it considers both speed and acceleration factors [14]. 

Studies that assessed the performance considering only the speed category have underestimated the amount of high-intensity activity performed by players. Indeed, when expressed as metabolic power, Gaudino et al., (2013) indicated that the MD showed a higher volume of high-intensity activity compared to attackers [48]. Our study confirms previous findings showing that midfielders showed higher metabolic power during an OM compared to other playing positions.

Specifically, our results show that the playing positions that recorded the highest MP average values were the WM, followed by the MD and ES, whereas the FB and CB were the lowest. In line with our results, Manzi et al., (2022) [63] reported that central backs covered less high-metabolic power distance and performed lower power events than players in the other playing positions, a finding likely related to the tactical role. Furthermore, midfielders covered both a substantial distance at high metabolic power and the largest number of power events compared to other playing positions.

This result could be supported by the fact that midfielders play in very dense central spaces of midfield that limit the performance of intense actions such as high-speed running and possibly increase the number of accelerations. It is worth noting that metabolic power increases because of speed or accelerations [14], consequently, since the MDs do not reach high distances at high intensity, their metabolic power increases more due to the numerous accelerations and decelerations. Furthermore, these results are in line with the results of a previous study showing that midfielders spend most of their time in medium and high-intensity activity during a match [39]. We also found that the MD and WM performed the highest distance per minute, whereas the CB and FB showed the lowest values. This result could be explained by the low recovery time and distance (i.e., TLMP and DLMP) among the actions detected in midfielders. 

In addition, supporting our results, a study reported that midfielders had greater power recovery than central backs and forwards and lower recovery time after power events than central back, full back, and forward players [63]. Our study provided a detailed analysis of metabolic power as different metabolic intensity thresholds were reported for each playing position. 

In line with previous findings, we found that the CB showed the lowest high, elevated, and maximum MP (expressed as both time and distance), whereas the MD and WM showed the highest THMP. However, considering the “intense” power thresholds (i.e., TEMP, TMMP, DEMP, and DMMP), the highest results were found in the ES and WM. In order to reach elevated and maximal metabolic power, players have to reach high speeds. This is possible by having space available as in the case of wide players such as the ES and WM. Indeed, Di Pampero et al., (2005) showed that the peak power output, of about 100 W/kg, is attained after about 0.5 s and that the average power over the first 4 s is on the order of 65 W/kg during 30 m running [33]. This means that players have to do a large number of intense long straight intense runs to achieve high and maximum metabolic power.

### 4.4. Strengths and Limitations of the Study

In this study, we considered cinematic, mechanical, and metabolic EL indicators that provide a complete overview of match performance and permit the evaluation of differences between playing positions. 

Furthermore, it is worth mentioning that a different tactical requirement, the score of the match, the quality of the opponent, and the fact that the present study is based on only four official matches considering only thirteen participants represent the limitations of this study.

### 4.5. Practical Implications 

In this study, we used a wearable inertial sensor (ISD technology) that provides several EL indicators useful to assess training and match performance in soccer. Compared to GPS, ISD can also be used for indoor sports without the need of coupling with external signals. 

Moreover, data sampling takes place differently. ISD measures movement in real time through limb swing, whereas GPS uses the Doppler effect of satellite signals which could increase the distortion of related signals. 

In summary, these devices can be useful for practitioners in assessing soccer players’ EL during training or competitions in order to prevent injuries and improve sports performance.

## 5. Conclusions

The present study provides a useful and novel insight into sprint and acceleration profiles of young Italian professional soccer players during OMs showing a difference in EL indicators between playing positions. In particular, considering cinematic, mechanical, and metabolic EL indicators, the ES, WM, and MD performed the highest level of physical performance during a match, whereas the CB and FB performed the lowest.

However, there is high variability in physical performance during a match between players of the same position [24] (e.g., FB) and a possible explanation for this observation could be that the match performance depends not only on playing position but also to some extent on individual players themselves [52]. 

This study reveals some new findings concerning the physical demand for each playing position during OMs. To design role-specific soccer training, coaches need a clear view of how different players and positions meet physical requirements. The principle of specificity (SAID) suggests that the training process should be designed to emphasize certain physical components to cope with match demands. Wide players exhibit greater sprinting distances than central players. Finally, as with sprinting, wide players seem to perform more accelerations than central players [12].

## Figures and Tables

**Figure 1 jfmk-08-00022-f001:**
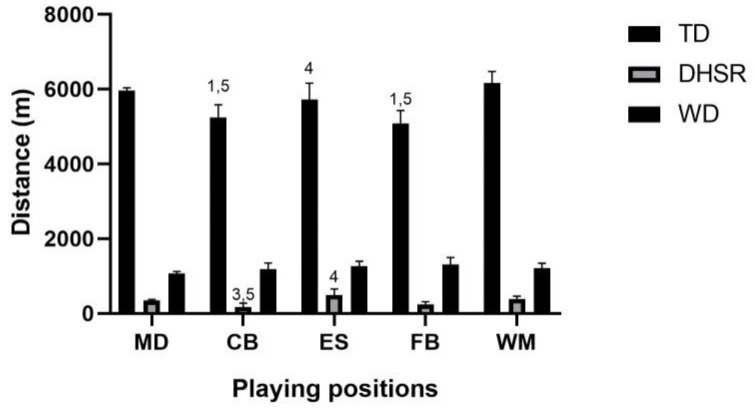
Playing position’s cinematic indicators (TD, DHSR, WD) during the first half of official matches. Legend: TD, Total Distance; MS, WD, Walking Distance; DHSR, Distance High-Speed Running; MD, midfielder; CB, central back, ES, external striker, FB, fullback, WM, wide midfielder; *p* < 0.05 for differences between playing positions (1 difference with midfielder; 3 difference with external striker; 4 difference with full back; 5 difference with wide midfielder).

**Figure 2 jfmk-08-00022-f002:**
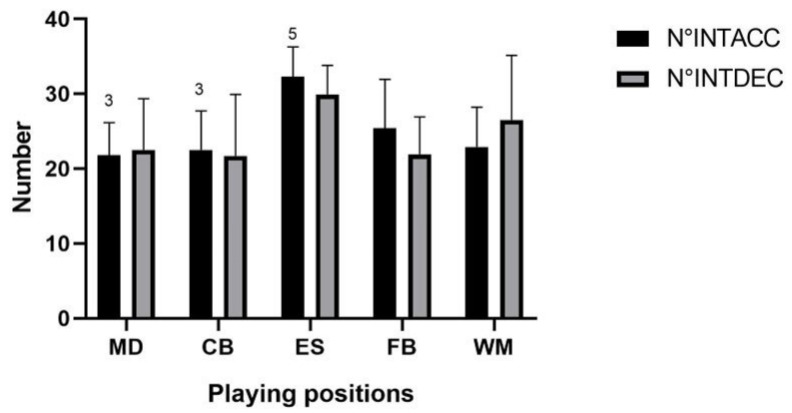
Playing position’s mechanics indicators (N°INTACC, N°INTDEC) during the first half of official matches. Legend: N°INTACC, Number of Intense Accelerations; N°INTDEC, Number of Intense Decelerations; *p* < 0.05 for differences between playing positions (3 difference with external striker; 5 difference with wide midfielder).

**Figure 3 jfmk-08-00022-f003:**
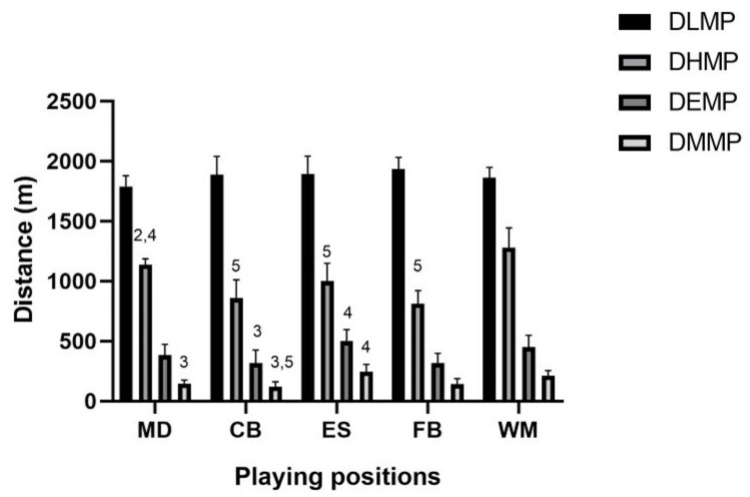
Playing position’s metabolic indicators (DLMP, DHMP, DEMP, DMMP) during the first half of official matches. Legend: DLMP, Distance Low Metabolic Power; DHMP, Distance High Metabolic Power; DEMP, Distance Elevated Metabolic Power; DMMP, Distance Max Metabolic Power; *p* < 0.05 for differences between playing positions (2 difference with central back; 3 difference with external striker; 4 difference with full back; 5 difference with wide midfielder).

**Table 1 jfmk-08-00022-t001:** Descriptions of the external load indicators.

Indicators	Type	Description (Unit of Measure)
TD	Cinematic/volume	Total distance covered (m)
MS *	Cinematic/intensity	Maximum speed reached (even for <1 s)
N°INTACC *	Mechanical/volume	Number of intense accelerations >2 m/s^2^
N°INTDEC *	Mechanical/volume	Number of intense decelerations >2 m/s^2^
TDA	Mechanical/volume	Distance traveled with positive acceleration (i.e., with speed increase) (m)
TDD	Mechanical/volume	Distance traveled with negative acceleration (i.e., with speed decrease) (m)
N°HSR *	Cinematic/volume	Number of high-intensity running at >20 km/h
WT	Cinematic/volume	Time spent in the various speed zones (<6 km/h) (s)
THSR *	Cinematic/volume	Time spent in the various speed zones (>20 km/h) (s)
WD	Cinematic/volume	Distance traveled in the various speed zones (<6 km/h) (m)
DHSR *	Cinematic/volume	Distance traveled in the various speed zones (>20 km/h) (m)
MP *	Metabolic/intensity	Metabolic Power (w·kg^−1^) was calculated by multiplying EC (in J·kg^−1^·m^−1^) by running speed (v; in m·s^−1^) at any given moment (i.e., every 0.2 s): P met = EC·v.In order to assess metabolic power, considering the energy expenditure and derived, the equation developed by di Prampero et al. [33] established on previously studies by Minetti et al. [34] and Osgnach et al. [14] was adopted. (Watt = w)
TLMP	Metabolic/volume	Time spent in various metabolic power zones (<10 w) (s)
THMP *	Metabolic/volume	Time spent in various metabolic power zones (20–35 w) (s)
TEMP *	Metabolic/volume	Time spent in various metabolic power zones (35–55 w) (s)
TMMP *	Metabolic/volume	Time spent in various metabolic power zones (>55 w) (s)
DLMP	Metabolic/volume	Distance traveled in the various metabolic power zones (<10 w) (m)
DHMP *	Metabolic/volume	Distance traveled in the various metabolic power zones (20–35 w) (m)
DEMP *	Metabolic/volume	Distance traveled in the various metabolic power zones (35–55 w) (m)
DMMP *	Metabolic/volume	Distance traveled in the various metabolic power zones (>55 w) (m)
N°CoDR *	Mechanical/volume	Number of direction changes to the right >30° and with speed >2 m/s
N°CoDL *	Mechanical/volume	Number of direction changes to the left >30° and with speed >2 m/s

Legend: TD, Total Distance; MS, Maximal Speed; N°INTACC, Number of Accelerations; N°INTDEC, Number of Decelerations; TDA, Total Distance Acceleration; TDD, Total Distance deceleration; N°HSR, Number of High-Speed Running; WT, Walking Time; THSR, Time High-Speed Running; WD, Walking Distance; DHSR, Distance High-Speed Running; MP, Metabolic Power; TLMP, Time Low Metabolic Power; THMP, Time High Metabolic Power; TEMP, Time Elevated Metabolic Power; TMMP, Time Max Metabolic Power; DLMP, Distance Low Metabolic Power; DHMP, Distance High Metabolic Power; DEMP, Distance Elevated Metabolic Power; DMMP, Distance Max Metabolic Power; N°CoDR, Number of Direction Changes to the Right; N°CoDL, Number of Direction Changes to the Left; * Intensity indicator.

**Table 2 jfmk-08-00022-t002:** Descriptive statistics of the external load indicators.

Cinematic									
	TD (m)	MS (km/h)	N°HSR (Total)	WT (s)	THSR (s)		WD (m)	DHSR (m)	
MEAN	5620	26.5	29.4	1451	54.7		1226	335	
SD	537	2.58	10.3	179	24.5		155	152	
MIN	4714	20.6	10	1106	13		987	76.5	
MAX	6557	30.8	58	1780	143		1539	880	
**Mechanical**									
	N°INTACC (total)	N°INTDEC (total)	TDA (m)	TDD (m)	N°CoDR (total)		N°CoDL (total)		
MEAN	25.5	24.9	3018	2582	143		139		
SD	6.35	7.09	309	239	15.9		31.6		
MIN	16	9	2466	2161	105		76		
MAX	37	38	3569	3071	173		220		
**Metabolic**									
	MP (w)	TLMP (s)	THMP (s)	TEMP (s)	TMMP (s)	DLMP (m)	DHMP (m)	DEMP (m)	DMMP (m)
MEAN	10.7	1739	250	80.3	30.8	1883	998	399	179
SD	1.12	139	46.3	21.5	10.6	120	207	119	67.5
MIN	8.7	1502	172	33	14	1668	628	136	77.4
MAX	13	2004	337	133	56	2197	1398	711	329

Legend: TD, Total Distance; MS, Maximal Speed; N°INTACC, Number of Intense Accelerations; N°INTDEC, Number of Intense Decelerations; TDA, Total Distance Acceleration; TDD, Total Distance deceleration; N **°** HSR, Number of High-Speed Running; WT, Walking Time; THSR, Time High-Speed Running; WD, Walking Distance; DHSR, Distance High-Speed Running; MP, Metabolic Power; TLMP, Time Low Metabolic Power; THMP, Time High Metabolic Power; TEMP, Time Elevated Metabolic Power; TMMP, Time Max Metabolic Power; DLMP, Distance Low Metabolic Power; DHMP, Distance High Metabolic Power; DEMP, Distance Elevated Metabolic Power; DMMP, Distance Max Metabolic Power; N°CoDR, Number of Change of Direction Right; N°CoDL, Number of Change of Direction Left; SD, standard deviation.

**Table 3 jfmk-08-00022-t003:** Descriptive statistics of the external load indicators for each official match.

	Cinematic									
	MATCH	TD (m)	MS (km/h)	N°HSR (Total)	WT (s)	THSR (s)	WD (m)	DHSR (m)		
MEAN	1	5708	26.8	29.8	1428	53.7	1248	328		
	2	5673	25.7	31.6	1330	60.3	1151	369		
	3	5456	26.8	26.6	1595	49.6	1325	303		
	4	5625	26.8	29.3	1469	54.8	1187	335		
SD	1	594	2.77	12.1	164	23.4	124	146		
	2	625	1.97	12.4	163	35.0	109	216		
	3	515	2.68	7.69	192	18.7	195	117		
	4	444	3.14	8.86	96.8	19.3	153	119		
MIN	1	4714	21.9	10	1177	13	1119	76.5		
	2	4728	22.4	14	1106	20	987	120		
	3	4936	23.3	14	1267	22	1079	131		
	4	4740	20.6	11	1318	13	1026	77.4		
MAX	1	6557	29.1	48	1705	81	1444	504		
	2	6544	28.4	58	1575	143	1343	880		
	3	6338	30.8	38	1780	82	1539	505		
	4	6162	30.2	41	1571	78	1409	477		
	**Mechanical**									
	MATCH	N°INTACC (total)	N°INTDEC (total)	TDA (m)	TDD (m)	N°CoDR (total)	N°CoDL (total)			
MEAN	1	26.3	27.6	3097	2592	148	139			
	2	24.8	23.1	3038	2616	138	146			
	3	24.4	22.9	2903	2532	134	125			
	4	26.4	25.9	3023	2581	150	145			
SD	1	5.74	7.11	345	265	12.5	30.9			
	2	6.89	7.94	356	275	18.3	33.3			
	3	5.60	4.97	305	215	15.8	39.2			
	4	7.93	7.97	229	228	12.8	21.8			
MIN	1	18	18	2485	2206	124	101			
	2	16	13	2466	2248	105	117			
	3	19	15	2581	2280	108	76			
	4	16	9	2557	2161	137	117			
MAX	1	35	36	3569	2970	164	198			
	2	35	38	3476	3071	162	220			
	3	35	28	3476	2845	153	184			
	4	37	35	3357	2904	173	188			
	**Metabolic**									
	MATCH	MP (w)	TLMP (s)	THMP (s)	TEMP (s)	TMMP (s)	DLMP (m)	DHMP (m)	DEMP (m)	DMMP (m)
MEAN	1	10.9	1711	257	85.6	31.0	1896	1028	423	181
	2	11.1	1646	252	80.9	32.0	1867	1009	408	183
	3	10.1	1850	236	71.9	28.9	1907	938	356	168
	4	10.6	1766	254	82.0	31.1	1862	1012	406	182
SD	1	1.22	137	50.1	19.8	11.6	114	223	111	77.1
	2	1.26	122	46.6	29.1	10.9	152	219	163	69.0
	3	0.964	138	52.6	11.1	11.8	116	219	59.8	76.4
	4	0.884	80.7	41.1	23.0	9.46	106	190	126	57.3
MIN	1	8.88	1511	181	56	15	1738	690	254	79.5
	2	9.15	1502	172	48	21	1668	628	213	106
	3	9.05	1659	178	50	15	1743	688	242	81.8
	4	8.70	1641	187	33	14	1763	685	136	77.4
MAX	1	12.6	1958	321	107	48	2060	1319	559	289
	2	13.0	1857	337	133	56	2197	1398	711	329
	3	11.8	2004	327	87	52	2068	1324	450	312
	4	11.6	1864	327	101	44	2069	1330	511	261

Legend: TD, Total Distance; MS, Maximal Speed; N°INTACC, Number of Intense Accelerations; N°INTDEC, Number of Intense Decelerations; TDA, Total Distance Acceleration; TDD, Total Distance Deceleration; N **°** HSR, Number of High-Speed Running; WT, Walking Time; THSR, Time High-Speed Running; WD, Walking Distance; DHSR, Distance High-Speed Running; MP, Metabolic Power; TLMP, Time Low Metabolic Power; THMP, Time High Metabolic Power; TEMP, Time Elevated Metabolic Power; TMMP, Time Max Metabolic Power; DLMP, Distance Low Metabolic Power; DHMP, Distance High Metabolic Power; DEMP, Distance Elevated Metabolic Power; DMMP, Distance Max Metabolic Power; N°CoDR, Number of Change of Direction Right; N°CoDL, Number of Change of Direction Left; SD, standard deviation.

**Table 4 jfmk-08-00022-t004:** External load indicator differences among playing positions (central back, external striker, full back, midfielder, wide midfielder).

	Midfielder (M ± SD)	Central Back (M ± SD)	External Striker (M ± SD)	Full Back (M ± SD)	Wide Midfielder (M ± SD)
Cinematic					
TD (m)	5963 ± 74	5240 ± 340 ^1,5^	5723 ± 438 ^4^	5087 ± 344 ^1,5^	6161 ± 316
MS (km/h)	25.7 ± 1.88	24.3 ± 2.66 ^3,5^	28.4 ± 1.34 ^4^	25 ± 2.61 ^5^	28.2 ± 1.31
N°HSR	29.8 ± 3.5	18.7 ± 7.89 ^3,5^	38.8 ± 8.88 ^4^	22.9 ± 6.53 ^5^	34.4 ± 7.03
WT (s)	1303 ± 78.5	1490 ± 60.5	1476 ± 205	1578 ± 187 ^5^	1345 ± 151
THSR (s)	56 ± 5.72	29.7 ± 15.6 ^3,5^	80.4 ± 26.3 ^4^	39.9 ± 12.4	62.1 ± 13.5
WD (m)	1071 ± 53.9	1191 ± 161	1260 ± 137	1308 ± 186	1214 ± 129
DHSR (m)	342 ± 36.3	179 ± 95.2 ^3,5^	494 ± 163 ^4^	242 ± 77.3	381 ± 84.6
Mechanical					
N°INTACC (total)	21.8 ± 4.35 ^3^	22.5 ± 5.21 ^3^	32.3 ± 3.99 ^5^	25.4 ± 6.55	22.9 ± 5.3
N°INTDEC (total)	22.5 ± 6.86	21.7 ± 8.21	29.9 ± 3.91	21.9 ± 5	26.5 ± 8.64
TDA (m)	3136 ± 59.1 ^3^	2818 ± 197 ^5^	3067 ± 251 ^4^	2732 ± 234 ^5^	3348 ± 202
TDD (m)	2810 ± 64.4 ^2,4^	2404 ± 159 ^3,5^	2639 ± 206 ^4^	2336 ± 117 ^5^	2789 ± 125
N°CoDR (total)	139 ± 5.74	143 ± 8.99	146 ± 18.9	131 ± 19.1 ^5^	153 ± 9.92
N°CoDL (total)	165 ± 6.08	152 ± 25.1	122 ± 25.1	149 ± 44.4	123 ± 15.4
Metabolic					
MP (w)	11.1 ± 0.233	9.9 ± 0.792 ^5^	11.1 ± 1.07 ^4^	9.68 ± 0.771 ^5^	11.7 ± 0.812
TLMP (s)	1625 ± 87.6 ^4^	1793 ± 54	1754 ± 135	1856 ± 128 ^5^	1626 ± 103
THMP (s)	286 ± 9.54 ^2,4^	221 ± 32.7 ^5^	245 ± 32.3 ^5^	209 ± 23.7 ^5^	301 ± 36.6
TEMP (s)	76.5 ± 17.7	67.2 ± 22.2 ^3^	97.4 ± 16.6 ^4^	66 ± 15.6	89.1 ± 19.2
TMMP (s)	25.3 ± 4.72 ^3^	21.8 ± 7.25 ^3,5^	42 ± 9.87 ^4^	25 ± 7.98	34.9 ± 5.67
DLMP (m)	1790 ± 90.8	1888 ± 153	1894 ± 148	1934 ± 98.3	1863 ± 86.5
DHMP (m)	1139 ± 46.8 ^2,4^	860 ± 151 ^5^	999 ± 150 ^5^	811 ± 110 ^5^	1281 ± 164
DEMP (m)	382 ± 92	316 ± 111 ^3^	502 ± 94.3 ^4^	316 ± 82.9	450 ± 98.2
DMMP (m)	144 ± 31.4 ^3^	120 ± 42.1 ^3,5^	247 ± 59.5 ^4^	140 ± 48.4	212 ± 43.9

Legend: TD, Total Distance; MS, Maximal Speed; N°INTACC, Number of Intense Accelerations; N°INTDEC, Number of Intense Decelerations; TDA, Total Distance Acceleration; TDD, Total Distance Deceleration; N°HSR, Number of High-Speed Running; WT, Walking Time; THSR, Time High-Speed Running; WD, Walking Distance; DHSR, Distance High-Speed Running; MP, Metabolic Power; TLMP, Time Low Metabolic Power; THMP, Time High Metabolic Power; TEMP, Time Elevated Metabolic Power; TMMP, Time Max Metabolic Power; DLMP, Distance Low Metabolic Power; DHMP, Distance High Metabolic Power; DEMP, Distance Elevated Metabolic Power; DMMP, Distance Max Metabolic Power; N°CoDR, Number of Change of Direction Right; N°CoDL, Number of Change of Direction Left. *p* < 0.05 for differences between playing positions (^1^ difference with midfielder; ^2^ difference with central back; ^3^ difference with external striker; ^4^ difference with full back; ^5^ difference with wide midfielder); M, mean; SD, standard deviation.

**Table 5 jfmk-08-00022-t005:** Highest and lowest external load indicator differences among playing positions (central back, external striker, full back, midfielder, wide midfielder).

Cinematic									
	TD (m)	MS (km/h)	N°HSR (Total)	WT (s)	THSR (s)	WD (m)	DHSR (m)		
Highest (M ± SD)	WM (6161 ± 316) MD (5963 ± 74)	ES (28.4 ± 1.34) WM (28.2 ± 1.31)	ES (38.8 ± 8.88 )WM (34.4 ± 7.03)	FB (1578 ± 187) CB (1490 ± 60.5)	ES (80.4 ± 26.3) WM (62.1 ± 13.5)	FB (1308 ± 186) ES (1260 ± 137)	ES (494 ± 163) WM (381 ± 84.6)		
Lowest (M ± SD)	FB (5087 ± 344) CB (5240 ± 340)	CB (24.3 ± 2.66) FB (25 ± 2.61)	CB (18.7 ± 7.89) FB (22.9 ± 6.53)	MD (1303 ± 78.5) WM (1345 ± 151)	CB (29.7 ± 15.6) FB (39.9 ± 12.4)	MD (1071 ± 53.9) CB (1191 ± 161)	CB (179 ± 95.2) FB (242 ± 77.3)		
**Mechanical**									
	N°INTACC (total)	N°INTDEC (total)	TDA (m)	TDD (m)	N°CoDR (total)	N°CoDL (total)			
Highest (M ± SD)	ES (32.3 ± 3.99) FB (25.4 ± 6.55)	ES (29.9 ± 3.91) WM (26.5 ± 8.64)	WM (3348 ± 202) MD (3136 ± 59.1)	MD (2810 ± 64.4) WM (2789 ± 125)	WM (153 ± 9.92) ES (146 ± 18.9)	MD (165 ± 6.08) CB (152 ± 25.1)			
Lowest (M ± SD)	MD (21.8 ± 4.35) CB (22.5 ± 5.21)	CB (21.7 ± 8.21) FB (21.9 ± 5)	FB (2732 ± 234) CB (2818 ± 197)	FB (2336 ± 117) CB (2404 ± 159)	FB (131 ± 19.1) MD (139 ± 5.74)	ES (122 ± 25.1) WM (123 ± 15.4)			
**Metabolic**									
	MP (w)	TLMP (s)	THMP (s)	TEMP (s)	TMMP (s)	DLMP (m)	DHMP (m)	DEMP (m)	DMMP (m)
Highest (M ± SD)	WM (11.7 ± 0.812) MD (11.1 ± 0.233)	FB (1856 ± 128) CB (1793 ± 54)	WM (301 ± 36.6) MD (286 ± 9.54)	ES (97.4 ± 16.6) WM (89.1 ± 19.2)	ES (42 ± 9.87) WM (34.9 ± 5.67)	FB (1934 ± 98.3) ES (1894 ± 148)	WM (1281 ± 164) MD (1139 ± 46.8)	ES (502 ± 94.3) WM (450 ± 98.2)	ES (247 ± 59.5) WM (212 ± 43.9)
Lowest (M ± SD)	FB (9.68 ± 0.771) CB (9.9 ± 0.792)	MD (1625 ± 87.6) WM (1626 ± 103)	FB (209 ± 23.7) CB (221 ± 32.7)	FB (66 ± 15.6) CB (67.2 ± 22.2)	CB (21.8 ± 7.25^3^) FB (25 ± 7.98)	MD (1790 ± 90.8) WM (1863 ± 86.5)	FB (811 ± 110) CB (860 ± 151)	FB (316 ± 82.9) CB (316 ± 111)	CB (120 ± 42.1) FB (140 ± 48.4)

Legend: TD, Total Distance; MS, Maximal Speed; N°INTACC, Number of Intense Accelerations; N°INTDEC, Number of Intense Decelerations; TDA, Total Distance Acceleration; TDD, Total Distance Deceleration; N°HSR, Number of High-Speed Running; WT, Walking Time; THSR, Time High-Speed Running; WD, Walking Distance; DHSR, Distance High-Speed Running; MP, Metabolic Power; TLMP, Time Low Metabolic Power; THMP, Time High Metabolic Power; TEMP, Time Elevated Metabolic Power; TMMP, Time Max Metabolic Power; DLMP, Distance Low Metabolic Power; DHMP, Distance High Metabolic Power; DEMP, Distance Elevated Metabolic Power; DMMP, Distance Max Metabolic Power; N°CoDR, Number of Change of Direction Right; N°CoDL, Number of Change of Direction Left; M, mean; SD, standard deviation.

**Table 6 jfmk-08-00022-t006:** One-way ANOVA test results.

Indicators	F	df1	df2	*p*
TD	16.59	4	13.9	<0.001
MS	5.54	4	11.8	0.009
N°INTACC	6.22	4	12.5	0.005
N°INTDEC	3.43	4	11.7	0.045
TDA	9.99	4	14.2	<0.001
TDD	23.29	4	13.7	<0.001
N°HSR	6.94	4	13.7	0.003
WT	5.21	4	13.1	0.01
THSR	7.82	4	14	0.002
WD	4.76	4	14	0.012
DHSR	7.89	4	14	0.002
MP	8.83	4	14.2	<0.001
TLMP	6.22	4	12.6	0.005
THMP	18.75	4	14.1	<0.001
TEMP	4.1	4	12.2	0.025
TMMP	7.04	4	13.1	0.003
DLMP	1.43	4	12.4	0.28
DHMP	15.51	4	14.1	<0.001
DEMP	5	4	12.3	0.013
DMMP	7.6	4	13.2	0.002
N°CoDR	2.93	4	13.9	0.06
N°CoDL	13.03	4	14	<0.001

Legend: TD, Total Distance; MS, Maximal Speed; N°INTACC, Number of Intense Accelerations; N°INTDEC, Number of Intense Decelerations; TDA, Total Distance Acceleration; TDD, Total Distance deceleration; N°HSR, Number of High-Speed Running; WT, Walking Time; THSR, Time High-Speed Running; WD, Walking Distance; DHSR, Distance High-Speed Running; MP, Metabolic Power; TLMP, Time Low Metabolic Power; THMP, Time High Metabolic Power; TEMP, Time Elevated Metabolic Power; TMMP, Time Max Metabolic Power; DLMP, Distance Low Metabolic Power; DHMP, Distance High Metabolic Power; DEMP, Distance Elevated Metabolic Power; DMMP, Distance Max Metabolic Power; N°CoDR, Number of Change of Direction Right; N°CoDL, Number of Change of Direction Left. *p* < 0.05 for differences between playing positions.

**Table 7 jfmk-08-00022-t007:** Multiple comparison test results.

Cinematic Indicators	
TD	* significant difference between MD and CB (*p* < 0.05), between CM and FB (*p* < 0.01), between CB and WM (*p* < 0.001), and between FB and WM (*p* < 0.001).
MS	* significant difference between CB and ES (*p* < 0.01), between CB and WM (*p* < 0.01), between ES and FB (*p* < 0.05), and between FB and WM (*p* < 0.05).
N°HSR	* significant difference between CB and ES (*p* < 0.001), between CB and WM (*p* < 0.01), between ES and FB (*p* < 0.01), and between FB and WM (*p* < 0.05).
WT	* significant difference between FB and WM (*p* < 0.05).
THSR	* significant between CB and ES (*p* < 0.001), between CB and WM (*p* < 0.05), and between ES and FB (*p* < 0.001).
WD	no significant difference between playing positions.
DHSR	* significant between CB and ES (*p* < 0.001), between CB and WM (*p* < 0.05), and between ES and FB (*p* < 0.001).
Mechanical Indicators	
N°INTACC	* significant difference between MD and ES (*p* < 0.05), between CB and ES (*p* < 0.05), and between ES and WM (*p* < 0.05).
N°INTDEC	no significant difference between playing positions.
TDA	* significant difference between FB and MD (*p* < 0.05), between CB and MD (*p* < 0.001), between FB and ES (*p* < 0.05), and between WM and FB (*p* < 0.001)
TDD	* significant difference between MD and CB (*p* < 0.01), between MD and FB (*p* < 0.001), between CB and ES (*p* < 0.05), between CB and WM (*p* < 0.001), between ES and FB (*p* < 0.01), and between FB and WM (*p* < 0.001).
N°CoDR	* significant difference between FB and WM (*p* < 0.05).
N°CoDL	no significant difference between playing positions.
Metabolic Indicators	
MP	* significant difference between CB and WM (*p* < 0.01), between ES and FB (*p* < 0.05), and between FB and WM (*p* < 0.001).
TLMP	* significant difference between MD and FB (*p* < 0.05), and between FB and WM (*p* < 0.001).
THMP	* significant difference between MD and CB (*p* < 0.05), between MD and FB (*p* < 0.01), between CB and WM (*p* < 0.001), between ES and WM (*p* < 0.01), and between FB and WM (*p* < 0.001).
TEMP	* significant difference between CB and ES (*p* < 0.05), and between ES and FB (*p* < 0.05).
TMMP	* significant difference between ES and MD (*p* < 0.01), between ES and CB (*p* < 0.001), between CB and WM (*p* < 0.05), and between ES and FB (*p* < 0.001).
DLMP	no significant difference between playing positions.
DHMP	* significant difference between MD and CB (*p* < 0.05), between MD and FB (*p* < 0.01), between CB and WM (*p* < 0.001), between ES and WM (*p* < 0.05), and between FB and WM (*p* < 0.001).
DEMP	* significant difference between CB and ES (*p* < 0.01), and between ES and FB (*p* < 0.001).
DMMP	* significant difference between MD and ES (*p* < 0.05), between CB and ES (*p* < 0.001), between CB and WM (*p* < 0.05), between ES and FB (*p* < 0.001), and between FB and WM (*p* < 0.05).

Legend: TD, Total Distance; MS, Maximal Speed; N°INTACC, Number of Intense Accelerations; N°INTDEC, Number of Intense Decelerations; TDA, Total Distance Acceleration; TDD, Total Distance deceleration; N°HSR, Number of High-Speed Running; WT, Walking Time; THSR, Time High-Speed Running; WD, Walking Distance; DHSR, Distance High-Speed Running; MP, Metabolic Power; TLMP, Time Low Metabolic Power; THMP, Time High Metabolic Power; TEMP, Time Elevated Metabolic Power; TMMP, Time Max Metabolic Power; DLMP, Distance Low Metabolic Power; DHMP, Distance High Metabolic Power; DEMP, Distance Elevated Metabolic Power; DMMP, Distance Max Metabolic Power; N°CoDR, Number of Change of Direction Right; N°CoDL, Number of Change of Direction Left. * *p* < 0.05 for differences between playing positions.

## Data Availability

The datasets generated during and/or analyzed during the current study are available from the corresponding author on reasonable request.

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
