# Peer review of "Match Load Physical Demands in U-19 Professional Soccer Players Assessed by a Wearable Inertial Sensor"

_jfmk, 2023, doi:10.3390/jfmk8010022_

Round 1

Reviewer 1 Report

The paper is greatly interesting for many fields, from biomechanics to sports psychology, and more.  But there are some items to improve to give it its full value. Notably, the uncertainties on the measurements of ISD, used for distances, speed, and more, but also for determination of changes of direction (for example, distances used in Tables 2 or 3 are measured with a +/- 0.5m, +/- 0.1m, or else, the speed with +/- x m/s?)  Also, Table 6, and some discussion items, use the term "significant" without any definition of what is significant, e.g. 1% or 10%, while it is key to understand).  These improvements would add perspective and further deepness to the relevant discussions therein.

Author Response

Dear Reviewer,

We appreciate the time and effort you have dedicated in order to improving the scientific soundness of our manuscript.

We have tried our best to follow your suggestions. We hope the work we have performed in this revised version results in an increase in the scientific soundness of the manuscript.

Below, a response to each comment is provided.

Thanks again.

REVIEWER 1

The paper is greatly interesting for many fields, from biomechanics to sports psychology, and more.  But there are some items to improve to give it its full value. Notably, the uncertainties on the measurements of ISD, used for distances, speed, and more, but also for determination of changes of direction (for example, distances used in Tables 2 or 3 are measured with a +/- 0.5m, +/- 0.1m, or else, the speed with +/- x m/s?)  Also, Table 6, and some discussion items, use the term "significant" without any definition of what is significant, e.g. 1% or 10%, while it is key to understand).  These improvements would add perspective and further deepness to the relevant discussions therein.

Dear Reviewer thank you very much for the comments. Significant changes, in accordance with your comments, are provided in the new version of the manuscript.

Moreover, we have improved the tables by adding the missing information as you suggested.

Reviewer 2 Report

The authors have sought to demonstrate differences in external loads using wearable technology between playing positions in soccer athletes. While this is an area that has recieved attention in the literature, the novel aspect of the sensor itself. At this time there are several aspects of the manuscript that need improvement before recommending acceptance for publication.

Introduction:

While the introduction provides a nice overview of the use of external loading metrics, I feel like this could be much more through discussion of positional differences not only at the professional, but high amateur (collegiate), and youth levels. Then adding the in the element of this being novel based on the technology used. I believe that the introduction lacks in terms of providing the background needed to full show the need for this investigation. Including information as to the issues with GPS and the need for other technologies for this information would add a lot of strength to this manuscript.

Methods:

How many players were in each group? With only having 13 participants in total this may be a large issue.

Was the data averaged over all the matches?

What was the means across each match? When discussing positional differences espically across multiple matches it is important to recognize that the demands of a given match may dicate the values. Thus, was this a trend across all matches or did one match impact the findings. This is data that needs to be reported in some way.

Results:

All Tables need to have girdlines removed and should be reformatted to have consisentecny.

Discussion:

Similar to the introduction there is a solid body of literature discussing external loading in soccer. I think again the highlight here needs to be the use of the ISD as a tool and how similar is this to GPS based data.

A further discussion into the use of ISD for metabolic I believe maybe warranted as well. While I understand the premise, it still may be a stretch from a prediction of metabolic loading from acceleration data. Additionally,, I would agrue that metabolic loads are actual internal not external and should be excluded from this study entirely. With that being said the agruement to include this and the calculations behind it should be discussed more throughly.

Author Response

Dear Reviewer,

We appreciate the time and effort you have dedicated in order to improving the scientific soundness of our manuscript.

We have tried our best to follow your suggestions. We hope the work we have performed in this revised version results in an increase in the scientific soundness of the manuscript.

Below, a response to each comment is provided.

Thanks again.

REVIEWER 2

The authors have sought to demonstrate differences in external loads using wearable technology between playing positions in soccer athletes. While this is an area that has received attention in the literature, the novel aspect of the sensor itself. At this time there are several aspects of the manuscript that need improvement before recommending acceptance for publication.

Dear Reviewer thank you very much for the comments. We followed the comments in order to improve the quality of the study.

Introduction:

While the introduction provides a nice overview of the use of external loading metrics, I feel like this could be much more through discussion of positional differences not only at the professional, but high amateur (collegiate), and youth levels. Then adding the in the element of this being novel based on the technology used. I believe that the introduction lacks in terms of providing the background needed to full show the need for this investigation. Including information as to the issues with GPS and the need for other technologies for this information would add a lot of strength to this manuscript.

Dear reviewer, thank you for the comments.

We have provided greater information concerning the levels of athletes (professional athletes, youth athletes, and amateurs).

According to our study, statistically significant differences were found, in external load indicators between playing positions, both in amateur athletes and in young athletes. Lines 73-90: we added the sentences: “…………in elite soccer players [21-25]. Moreover, significant differences were found between the various playing positions for all measures of external load in the amateur soccer. Authors reported that central midfielders covered the longest distance during a match, which is in line with recent literature followed by the forwards, the full-backs, and the wide midfielders. In accordance with adult professional soccer players physical performance is affected by playing position in elite youth players. In male elite soccer players (818 years), center backs covered the shortest and wide midfielders the longest high intensity running and sprinting distances. Also, center defenders covered the shortest very high-speed ( 19.8 km · h1) and sprint ( 25.2 km · h 1) distances, while wide players and center forwards covered the longest distances in these speed zones in young soccer players (mean age 16.0 years), which is in line with other studies.

It is worth noting that the GPS may have some signal issues due to adverse weather conditions or being used indoors compared to ISD. In fact, being this technology based on (inertial) movement, allows to evaluate the external load considering the same parameters relating to distance, speed and metabolic power with greater precision than GPS at 10 Hz. Indeed, the better applicability of ISD technology is due to its small size, lower cost, and the possibility of using the device indoors, avoiding possible connection problems between GPS and satellite.

  We have also tried to make it clearer the reason for choosing an Inertial Sensors Devices (ISD), rather than using a simple GPS device.

Methods:

How many players were in each group? With only having 13 participants in total this may be a large issue.

Was the data averaged over all the matches?

What was the means across each match? When discussing positional differences espically across multiple matches it is important to recognize that the demands of a given match may dicate the values. Thus, was this a trend across all matches or did one match impact the findings. This is data that needs to be reported in some way.

Dear reviewer, thank you for the comments.

The number of athletes included in the study is 13 because it represents the number of athletes who took part in the individual match played. We agree that 13 participants could be few, although being an observational study and not RTC, it is important to refer to the number of observations for each match. For clarity, we have included within the limits of the study that the sample is not very large.

The values entered are an average of the parameters detected in the four matches analysed, however, as suggested by your comment, we have inserted a table (table 3), which shows the values of the single parameters analysed for each match. In detail, the mean, the standard deviation, the minimum value, and the maximum value of each parameter analyzed were provided for each single match.

In the methods sections we added the players number of each playing position as you suggested.

Results:

All Tables need to have girdlines removed and should be reformatted to have consisentecny.

Dear reviewer, thank you for the comments.

As suggested by your comment, all gridlines have been removed, and the tables formatted to make them consistent.

Discussion:

Similar to the introduction there is a solid body of literature discussing external loading in soccer. I think again the highlight here needs to be the use of the ISD as a tool and how similar is this to GPS based data.

A further discussion into the use of ISD for metabolic I believe maybe warranted as well. While I understand the premise, it still may be a stretch from a prediction of metabolic loading from acceleration data. Additionally, I would argue that metabolic loads are actual internal not external and should be excluded from this study entirely. With that being said the agreement to include this and the calculations behind it should be discussed more thoroughly.

Dear reviewer, thank you for the comments.

Based on your suggestion, we have made the discussion more detailed about using ISD as a tool and how similar it is to GPS-based data, including the possible limiting factors in using GPS.

  • Lines 210-217. We added the sentences:” This technology allows to assess external load considering several indicators similar to GPS avoiding signal issue and connections problems. Moreover, Coutts et al. have demonstrated that GPS devices are reliable in assessing total distance and peak speed during high-intensity intermittent exercise but are less reliable for high-intensity activities such as accelerations. In contrast, the ISD device tracks activity information using inertial technology that captures acceleration and rotation data in real time at a rate of 100Hz per channel, resulting in more accuracy for detecting acceleration and high-intensity effort than GPS.

  • Concerning the “metabolic load” question, we have provided below some explanation to better clarify the concept behind the use of these indicators.

Published studies have referred to running speed to determine various high-intensity thresholds in soccer. In 2010, Osgnach and colleagues have revolutionized the concept of intensity in soccer by realizing that intensity is underestimated considering only running speed indicators. In fact, football is characterized by intermittent activities in which intense actions are repeated, including accelerations performed even at low speed. The authors have determined the energy cost of accelerating running (obviously higher than the energy cost at constant speed) through a theoretical-mathematical model. In this way, authors have estimated the energy cost of accelerations and a derived parameter i.e., metabolic power. While we agree that metabolic load is an “internal” (physiological) indicator, metabolic power has been considered by all published studies as external load parameter. Indeed, as reported in our studies (table 1): “Metabolic Power (w · kg−1) was calculated by multiplying energetic cost of accelerations (EC, in J · kg−1 · m−1) by running speed (v; in m · s−1) at any given moment (i.e., every 0.2 s): P met = EC*v. In order to assess metabolic power, considering the energy expenditure and derived, the equation developed by di Prampero et al. [29] established on previously studies by Minetti et al. [30] and Osgnach et al.[14] was adopted. (Watt=w)”.

Metabolic power can be modified when both running speed and/or EC of accelerations increase or decrease. It is change instantaneously. Moreover, the higher the sampling frequency, the greater and more accurate will be the changes in metabolic power taken into consideration by the device. For example, a 100Hz device will be more accurate in detecting accelerations during performance than GPS 10 or 20 Hz.

This parameter allows to assess the activities “intensity” including both speed and accelerations contribute.

Although we believe that metabolic load can be referred to internal load, we have included “metabolic power” that represent an external indicator. We know that metabolic power presents some limitation, as reported by Osgnach study but to date would be impossible to directly measure the “real” metabolic energy expenditure cost during soccer performance (it would be necessary to use the K4 metabolimeter together with GPS or ISD devices). We believe that the use of this parameter is highly suitable for monitoring player performance (external load) as supported by relevant published studies.

Thank you very much for the opportunity to explain this concept.

Round 2

Reviewer 2 Report

The authors have addressed the concerns raised during the initial review and provided an improved manuscript through the addition of more detail. Thank you for providing me with the information regarding the inclusion of the metabolic loads which I understand. My concern still lays in the title of the manuscript only discusses external loads and the metabolic are estimations and not actually measured (though they could be). I would maybe adjust the title if you choose to continue to include these metrics.

Author Response

Dear Reviewer,

Thank you for your comment.

According to your suggestion we changed the title in: “Match-load physical demands in U-19 professional soccer players assessed by a wearable inertial sensor”